# Photo-thermoresponsive polypyrrole-crosslinked single-chain nanoparticles for phototherapy
Justus Friedrich Thümmler [1], Farzin Ghane Golmohamadi [2], Daniel Schöffmann[2], Jan Laufer [2], Henrike Lucas [3], Julia Kollan [3], Karsten Mäder [3] & Wolfgang Hubertus Binder [1] ✉

Irradiating a chromophore allows cancer diagnostics by photoacoustic (PA) imaging, but also causes transformation of light into thermal energy and so enables therapy by photothermal effects. Useful chromophores for photothermal therapy (PTT), such as nanoconfined oligomers and polymers, should not only display an excellent light-to-heat efficiency, but in addition must display biocompatibility and good water-solubility. We in this study have designed water-dispersible, nanocaged polypyrroles (PPy), embedded into single-chain nanoparticles (SCNP, sized 6.8 – 8.9 nm) displaying thermoresponsivity, so reaching largely increased PTT-effects. Such encaged PPy-SCNPs allow an efficient photothermal heat conversion reaching temperatures up to 85°C. In a parallelized 96-well-plate-design the PPy-SCNPs can effect an almost complete death of illuminated (cancer) cells at already low concentrations (0.001 mg/mL) with low radiant fluxes. The thermoresponsivity of the SCNPs, surrounding the PPy-chains, then responds to the photothermal heat, so creating a unique self-amplifying effect for photothermal therapy and photoacoustic imaging.

Transformation of light into thermal energy is a central process in many catalytic and biomedical processes, among them photothermal conversion, wherein absorbed light is most efficiently transformed in heat by radiationless-decay of the absorbed photons[1,2]. A large variety of different materials can efficiently transform light into heat[3,4], such as plasmonic metallic nanostructures, nonplasmonic semiconductors[1], or organic materials[5], displaying extended π-systems with an often sophisticated molecular (twisted) design[6] or specifically aggregated states[7–9]. The formation of specific nanoparticulate organic materials (usually sized 40 - 200 nm)[8,9] has led to interest in organic dyes, resulting in enhanced efficiencies of the photothermal conversion processes[10] in vivo as well as in vitro. Based on their efficiency in photothermal conversion processes (nano-) materials are therefore relevant in photothermal tumor therapies (PTT)[11–13], photoacoustic (PA) imaging[8,14] and chemical catalysis[3]. PTT uses the photothermal effect of near-infrared (NIR) chromophores, able to locally generate heat after irradiation with NIR light to enable cell death of the tissues containing the photothermal drug[5–7]. Photothermal conversion is further used for photoacoustic (PA) imaging[8,10,15], allowing deep tissue imaging with penetration depth in the cm-range with tens of microns spatial resolution, advantageous over other imaging methods in specificity and resolution. Both methods can be combined into a theranostic approach, using the NIR chromophores first for diagnostic photoacoustic imaging and subsequently for therapeutic photothermal therapy[9]. Especially polypyrrole (PPy) as an organic semiconductor has been proven to very effectively convert excitation energy into heat, applicable for photothermal therapy and photoacoustic imaging applications[16–25]. As native PPy is not soluble in most solvents (including water), it must be applied in colloidal form, often using water-dispersible PPy nanoparticles by oxidative polymerization of pyrrole in the presence of stabilizing polymers, e.g., poly vinylacetate or poly ethylenimine with diameters of 10 – 130 nm or larger, depending on the ratio of PPy to stabilizing polymer[17–19,21]. PPy can be also be generated as nanosheets with lateral sizes of 100–200 nm and heights of 2 nm, displaying a strong absorption behavior especially in the second NIR window[16].

In this study we embed in situ generated PPy chromophores into ultra-small single-chain nanoparticles (SCNPs) with diameters < 10 nm, reaching both, an excellent, ultrasmall-sized and water soluble PPy-chromophore for photoacoustic/photothermal effects, coupled to a thermally responsive system, wherein the photothermal effect is linked to a singular response of the particle it is embedded into. For our concept, we rely on SCNPs, polymeric nanostructures generated by intramolecularly collapsing and

[1]Institute of Chemistry, Faculty of Natural Science II, Martin Luther University Halle-Wittenberg, von-Danckelmann-Platz 4, Halle, Saale, D-06120, Germany. [2]Institute of Physics, Faculty of Natural Science II, Martin Luther University Halle-Wittenberg, von-Danckelmann-Platz 3, Halle, Saale, D-06120, Germany. [3]Institute of Pharmacy, Faculty of Natural Science I, Martin Luther University Halle-Wittenberg, Kurt-Mothes-Straße 3, Halle, Saale, D-06120, Germany. ✉e-mail: wolfgang.binder@chemie.uni-halle.de

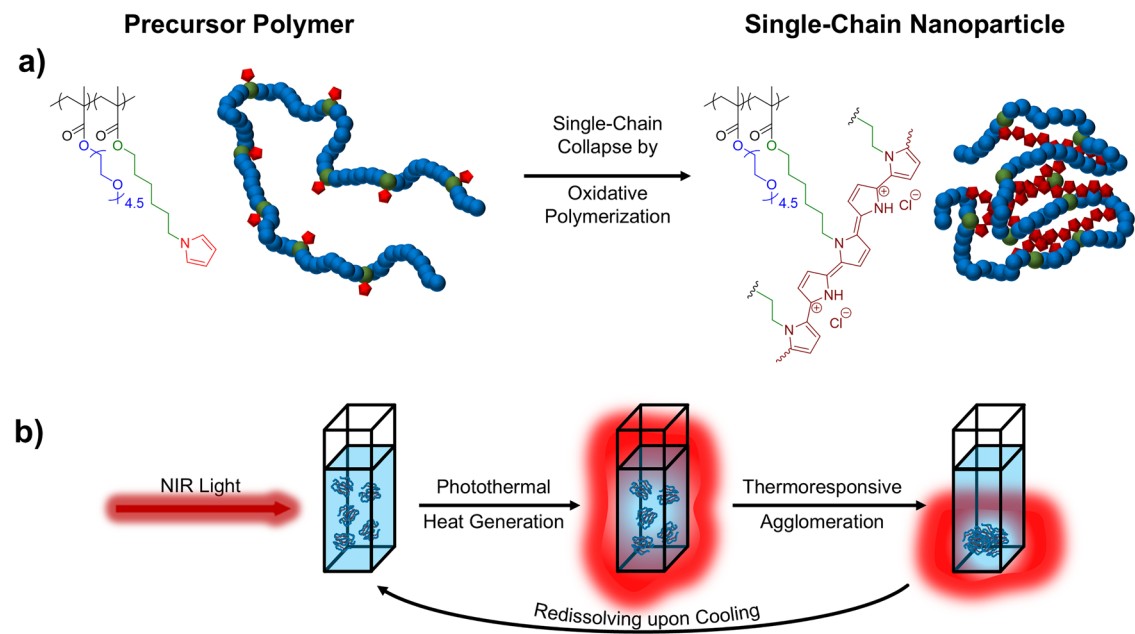

**Scheme 1 |** Concept of the photo-thermoresponsive SCNPs. **a** Illustration of the generation of SCNPs *via* oxidative polymerization of pyrrole sidechains, and (**b**) their use in photothermal therapy. The OEGMA-based SCNPs show LCST-type behavior resulting in the formation of water-insoluble agglomerates upon irradiation with NIR light.

**Table 1 | Summarized data of the precursor polymer and the SCNPs: average number of pyrrole units per chain, PPy content in the SCNP, apparent number average molecular weight $M_n^{app}$, degree of compaction $G$ after single-chain collapse, polydispersity index Đ, hydrodynamic diameter $D_h$, cloud point temperature $T_{cp}$, and mass extinction coefficient $\varepsilon$**

| | average number of Py units per chain | PPy content (µg mg⁻¹) | $M_n^{app\ a}$ (kDa) | $G^b$ (%) | Đ | $D_h^c$ (nm) | $T_{cp}^d$ (°C) | $\varepsilon$ (722 nm)ᵉ (mL mg⁻¹ cm⁻¹) | $\varepsilon$ (815 nm)ᵉ (mL mg⁻¹ cm⁻¹) |
|---|---|---|---|---|---|---|---|---|---|
| Polymer | 8.5 | — | 25.0 | — | 1.47 | 9.42 ± 0.05 | 49.5 | — | — |
| SCNP₀ | 8.5 | 22.8 | 14.2 | 43.2 | 1.25 | 8.2 ± 0.1 | 49.4 | 0.14 | 0.18 |
| SCNP₂ | 25.5 | 65.4 | 14.2 | 43.2 | 1.38 | 8.4 ± 0.5 | 50.3 | 1.53 | 1.04 |
| SCNP₅ | 51.0 | 122.9 | 13.1 | 47.6 | 1.30 | 8.9 ± 0.6 | 49.5 | 3.51 | 2.52 |
| SCNP₁₀ | 93.5 | 204.2 | 11.9 | 52.4 | 1.34 | 6.8 ± 0.6 | 52.9 | 15.31 | 14.25 |

ᵃ Measured by GPC in THF.

ᵇ Compaction factor $G$ calculated with $G = 100\ (M_{n,Polymer} - M_{n,SCNP}^{app})/M_{n,Polymer}$.

ᶜ Measured by DOSY-NMR in D₂O at concentrations of 1 mg mL⁻¹, calculated using $D_h = (k_B \cdot T) / (3 \cdot \pi \cdot \eta \cdot D)$.

ᵈ Measured by turbidimetry at concentrations of 1 mg mL⁻¹, $T_{cp} = T$ at 90% transmittance, see Fig. 1d. Error is ±0.2 °C, after statistical analysis ᵉ Based on the total mass of pyrrole in the system.

crosslinking single polymer chains. By carefully designing the precursor polymers and the reaction conditions by which the SCNPs are synthesized, internal nanostructures are formed[26–28], resembling pockets in proteins and enzymes[27–31], wherein the PPy is embedded in situ. Such internal "compartments" can form local environments for embedded molecules that have been exploited for enzyme-like catalysis[32–36] and biomedicine[37–40]. Examples for such compartmentalized SCNPs include crosslinked block copolymers with vinyl groups in both blocks using a one-pot metathesis polymerization in only one of the blocks[41], resulting in phase-separated Janus-shaped SCNPs; or benzene-1,3,5-tricarboxamides (BTA) containing polymers crosslinked by helical hydrogen bonding stacks inside the SCNPs[42–48]. Compartments inside such SCNPs (sized ~ 1 nm) based on single-chain collapsed copolymers have been demonstrated by us recently, containing a specific number of NIR dye molecules (up to ~10) in one compartment, in turn enhancing the PA-effect of the embedded dyes upon irradiation with an NIR-laser[27,28,37].

We therefore considered the embedding of the PPy chromophore directly into a compartment of a SCNP as an advantageous method to exploit the PTT efficiency inside such nanoparticles. The here presented combination of the photothermal effect of the PPy and the thermo-responsivity of the SCNP shell is supposed to effect a direct conversion of the absorbed energy inside the SCNP into a thermal effect at the interface of the

nanoparticles (see Scheme 1). Polymerization of pyrrole inside the SCNP serves two purposes: it firstly enables intramolecular crosslinking of individual polymer chains, resulting in PPy embedded into SCNPs of less than 10 nm in diameter, so probing whether such small PPy-particles can be effective in photothermal effects. Second, embedding the PPy-polymer directly inside a poly(oligo ethylene glycol) methacrylate (OEGMA)-based SCNP protects the chromophore from photooxidation, a major cause of reduced PTT and PA effects after multiple irradiations, but also should allow transfer of energy to the surrounding, thermoresponsive lower-critical-solution-temperature (LCST) type polymer-segments, so inducing a responsive effect by volume changes, potentially allowing enrichment of the SCNPs by NIR-excitation at the irradiated region.

The chemical design of the here presented PPy-crosslinked SCNPs is based on the direct in-situ-incorporation of PPy into a thermoresponsive SCNP via oxidative polymerization, tuning the amount of PPy formed inside the SCNP. This enables (i) an increased aqueous solubility of PPy and biocompatibility by the oligo ethylene glycol (OEG)-based SCNP-shell; (ii) encapsulation of the chromophore inside the ultra-small SCNPs for enhanced permeation and retention (EPR)[49–52]; and (iii) embedding a thermoresponsive lower critical solution temperature (LCST) behavior via the thermoresponsive OEG-side chains in the shell. Thereby we expect a self-amplifying photo-thermoresponsive mechanism for the PPy-SCNPs,

**Fig. 1 | Proof of successful single-chain collapse.**
**a** [13]C-NMR spectra of 6-pyrroylhexanol, 6-pyrroylhexyl methacrylate, the precursor polymer, and SCNP2, **b** GPC traces of the precursor polymer and the resulting SCNPs. **c** AFM height profile of SCNP5.

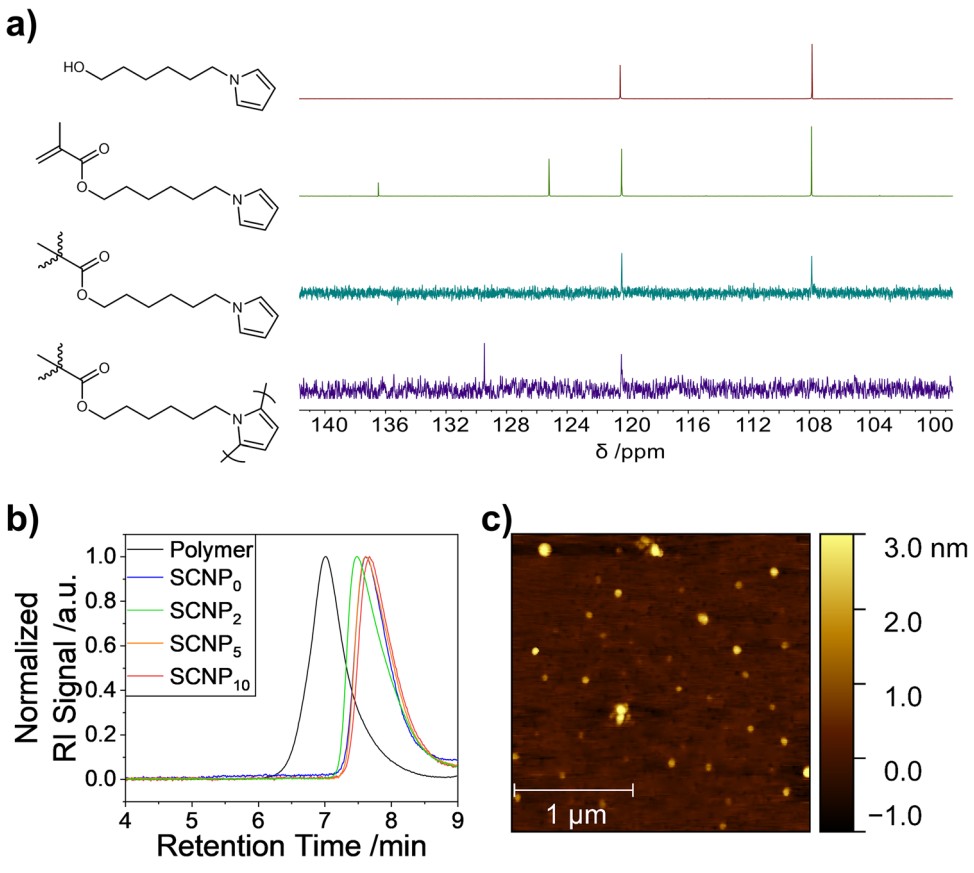

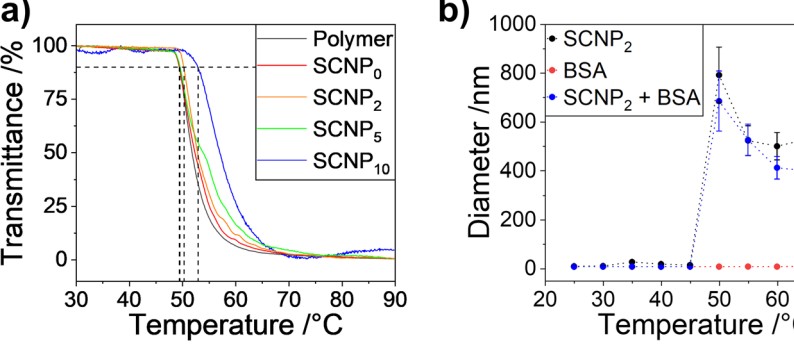

**Fig. 2 | Thermoresponsivity of the SCNPs.**
**a** Turbidimetry measurements of the precursor polymer and the resulting SCNPs in water at 1 mg/mL. **b** Hydrodynamic diameter of SCNP2 (1 mg/mL) and 10% BSA solution in phosphate buffered saline measured by temperature dependent dynamic light scattering.

reversibly agglomerating and therefore enriching at the site of the photo-thermal effect, as depicted in Scheme 1b.

## Results and Discussion

We started the design with the synthesis of a precursor polymer, containing a small amount of the pyrrole-units directly connected to a long polymer chain ($M_n > 20$ kDa), that was subsequently collapsed into a SCNP. The bound, small number of pyrrole-units served as an initiator for the subsequent oxidative polymerization of further added pyrrole to crosslink the precursor polymer and so form the PPy directly inside the SCNPs. A pyrrole-containing methacrylate monomer was synthesized *via* an $S_{N}2$ reaction on a 6-chlorohexanol spacer (see Figure S1 and S2), followed by subsequent esterification reaction with methacryloyl chloride yielding 6-pyrrolylhexyl methacrylate (PyrHexMA, see Figure S3 and S4). The precursor polymer for the single-chain collapse was accomplished via atom transfer radical copolymerization (ATRP) of the methacrylates in toluene using oligo ethylene glycol sidechains (OEGMA, $M_n = 300$ Da) to embed the desired water solubility, biocompatibility, and thermoresponsivity; together with PyrHexMA as crosslinker and the PPy as the NIR chromophore. The chosen OEG-side chains ($n = 4.5$) are projected to induce a thermal collapse around 50 °C, which is a temperature expected to be reached in the photothermal processes. The polymerization yielded the precursor polymer with a molecular weight of $M_n = 25.0$ kDa, $Đ = 1.47$, with a monomer composition of 90% OEGMA and 10% PyrHexMA, and a degree of polymerization of $n = 85$, as calculated from the [1]H-NMR spectrum (see Figure S5), respectively. Hence, the so obtained copolymer statistically contained 8.5 pyrrole units per chain.

Single-chain collapse reaction was performed by oxidative polymerization of the pyrrole units following the continuous addition protocol: an aqueous solution of the precursor polymer together with 0, 2, 5, and 10 equivalents of additional pyrrole with respect to the pyrrole in the polymer chain (total pyrrole units/chain are shown in Table 1) was added slowly to a

**Fig. 3 | Photophysical behavior of the SCNPs.**
**a** Absorption spectra of the SCNPs at concentrations of 0.2 mg(SCNP)/mL in water. **b** Lambert-Beer plots of the SCNPs at 815 nm (the plots which refer to the SCNP concentrations instead of PPy concentrations are depicted in Figure S10). **c** Photos of SCNP solutions at 1 mg(SCNP)/mL in water.

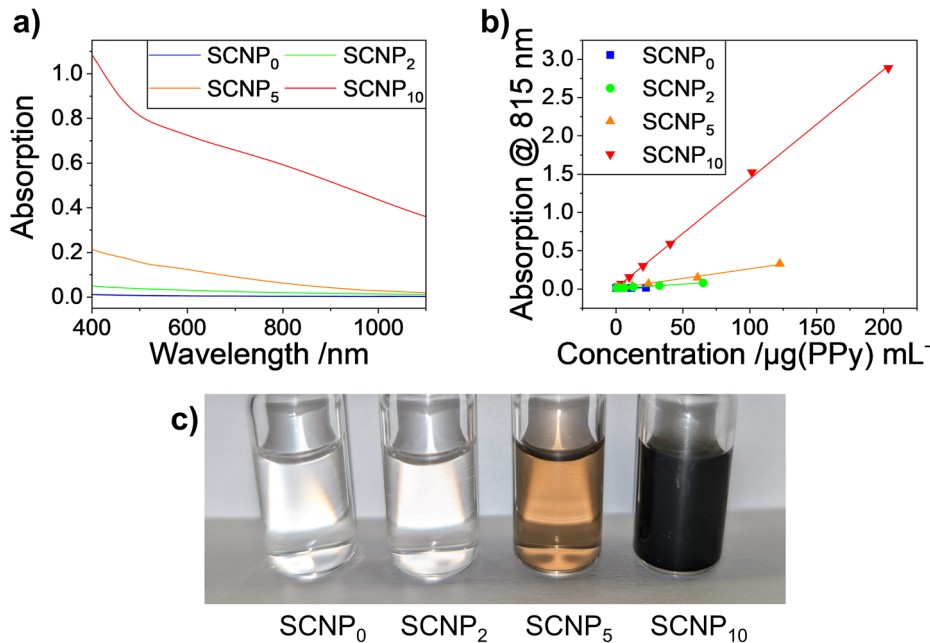

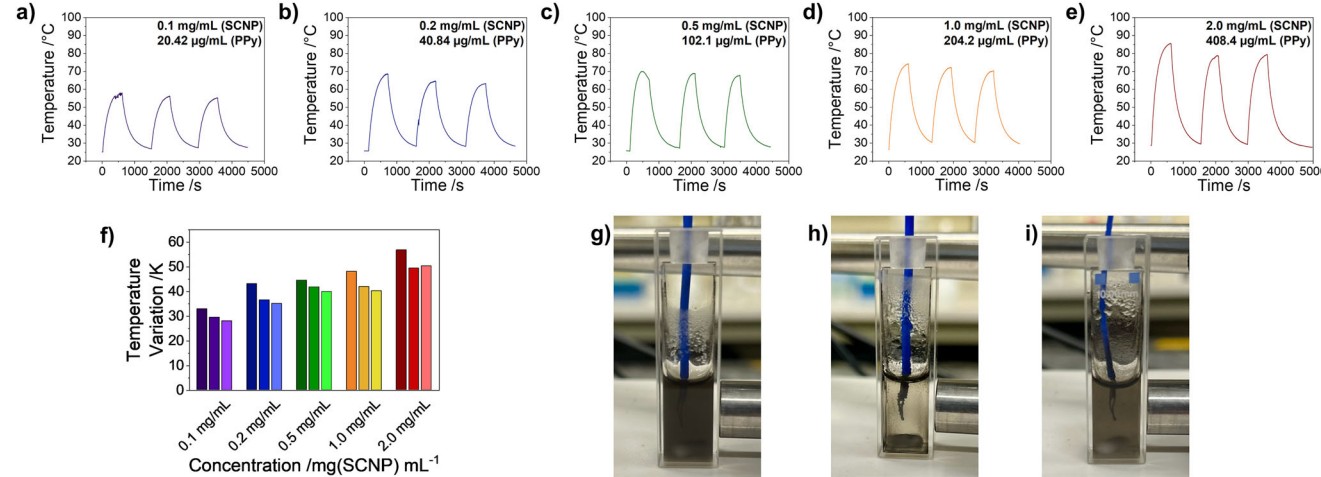

**Fig. 4 | Laser induced photothermal heating and cooling of PPy-containing SCNP$_{10}$ in water.** SCNP-concentrations were set to (**a**) 0.1 mg/mL, **b** 0.2 mg/mL, **c** 0.5 mg/mL, **d** 1.0 mg/mL, **e** 2.0 mg/mL. **f** Temperature variations after each irradiation cycle. Photos of the 0.5 mg/mL solution during the measurement: **g** before irradiation, **h** at the end of irradiation, **i** after cooling down to room temperature.

**Table 2 | Maximum temperatures $T_{max}$ and maximum temperature variation $T_{var, max}$ at different concentrations of SCNP$_{10}$ in water after the first irradiation cycles**

| SCNP$_{10}$ concentration (mg mL$^{-1}$) | PPy concentration (µg mL$^{-1}$) | $T_{max}$ (°C) | $T_{var, max}$[a] (K) |
|---|---|---|---|
| 0.1 | 20.42 | 57.8 | 32.9 |
| 0.2 | 40.84 | 68.5 | 43.1 |
| 0.5 | 102.1 | 69.8 | 44.5 |
| 1.0 | 204.2 | 74.1 | 48.1 |
| 2.0 | 408.4 | 85.4 | 56.8 |

[a]Calculated from the difference between the starting temperature and the maximum temperature $T_{max}$.

vigorously stirred solution of an excess of FeCl$_3$ in water as oxidizing agent to induce pyrrole-polymerization (PPy). After purification by extraction, filtration, and silica gel chromatography to remove eventual PPy not covalently bound to the SCNP, brown-to-black SCNPs were obtained with hydrodynamic diameter of 6.8–8.9 nm, as determined by DOSY-NMR spectroscopy (see Table 1), well soluble in water and in a variety of organic solvents, such as chloroform, dichloromethane, or THF. $^{13}$C-NMR spectroscopy proved the polymerization of the pyrrole sidechains as evidenced by the elimination of the signal at 107.8 ppm and the generation of a new signal at 129.5 ppm (see Fig. 1 and Figures S5-8). The successful syntheses of SCNPs with solely intramolecular crosslinks *via* pyrrole-pyrrole connections were proven by reduction in hydrodynamic diameters *via* GPC measurements in THF, indicating specific reduction in the apparent molecular weights ($M_n$^app^), indicative of a degree of compaction of $G = 40–50\%$ (see Table 1 and Fig. 1b). Compaction of the SCNPs increased with increasing amount of added free pyrrole during the collapse reactions, ranging from 43.2% without additional pyrrole (SCNP$_0$) to 52.4% with 10

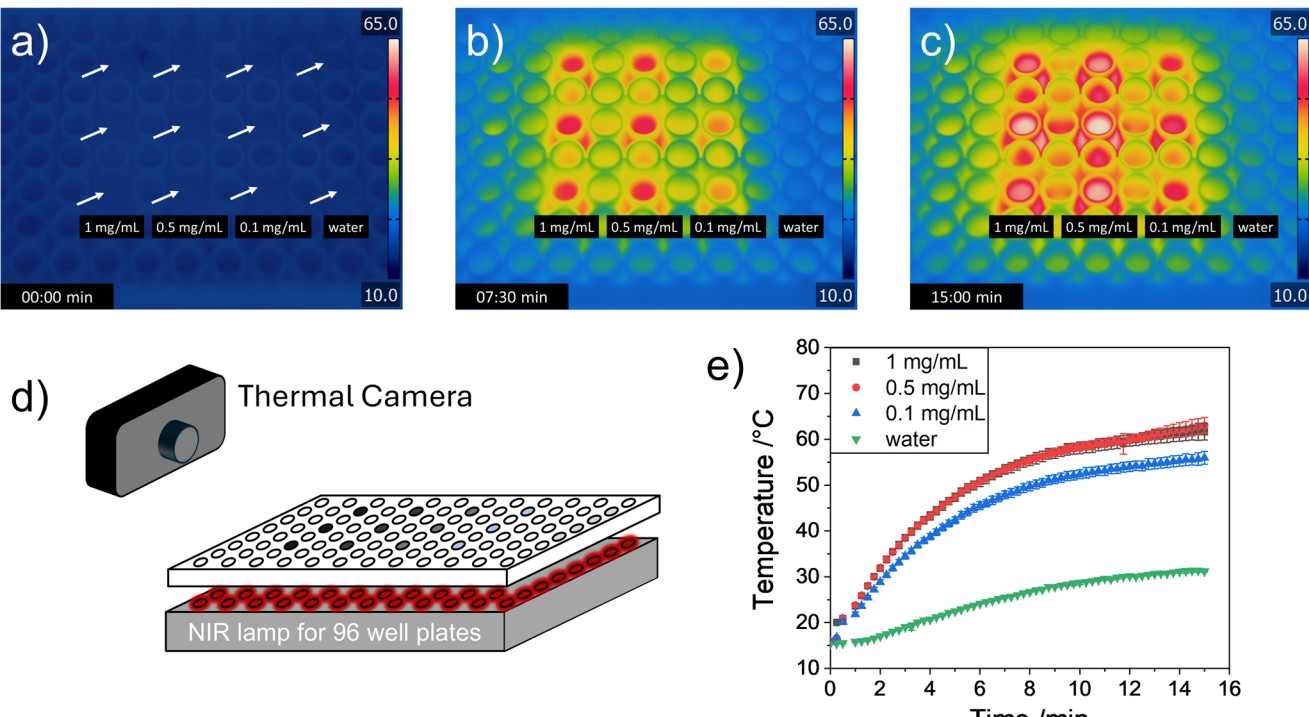

**Fig. 5 | LED induced photothermal heating and cooling of PPy-containing SCNP$_{10}$ in water.** Thermal pictures of the irradiated aqueous solutions of SCNP$_{10}$ (concentrations: 0 (water); 0.1; 0.5 and 1.0 mg/mL) using the well plate NIR-LED illumination mat (722 nm) with an irradiance of 0.84 W/cm² (**a**) before activating the LEDs (white arrows indicate position of the samples), **b** after 7.5 min, **c** after 15 min. **d** Schematical illustration of the measurement setup. **e** Averaged heating curves of the wells with different concentrations of SCNP$_{10}$ and water.

eq. of additional pyrrole (SCNP$_{10}$) with respect to the pyrrole in the precursor polymer chain, as shown in Table 1. Thus the additional pyrrole units caused the formation of longer, but also denser crosslinks of the PPy inside the SCNP by stronger segregation effects between the insoluble PPy core and the hydrophilic shell of the SCNP, and thus more compact SCNPs[53].

The compaction of the individual polymer chains after single-chain collapse could also be followed by atomic force microscopy (AFM, see Fig. 1c and S9). Excluding larger agglomerates of multiple molecules from the size calculation, the precursor polymer was found to show an average height of $5.7 \pm 1.7$ nm, while the collapsed and crosslinked SCNP$_5$ was found to display an average height of $2.6 \pm 0.4$ nm, hence, again proving a compaction of roughly 50%. We emphasize that the heights found by AFM measurements are significantly smaller than those calculated from DOSY-NMR measurements, caused by the loss of the globular structure of the solvated molecules when they are in a dry state and placed on a flat surface.

The SCNPs were designed thermoresponsive by their OEG sidechains embedded into the SCNP shell which cause LCST behavior in water. Thermally induced agglomeration was measured via turbidimetry revealing the cloud point temperatures $T_{cp}$ in the range of $T_{cp} \approx 50$ °C, as shown in Fig. 2a and Table 1. Below $T_{cp}$, the SCNPs are well dispersed with hydrodynamic diameters below 10 nm, (reversibly) collapsing into larger agglomerates of 500–800 nm above $T_{cp}$, as shown by temperature dependent hydrodynamic diameter from dynamic light scattering in Fig. 2b.

We further probed this effect under more realistic conditions, using a mixture of the SCNPs with the model protein BSA in phosphate buffer to simulate the behavior in the presence of plasma proteins by heating the solution above $T_{cp}$ (see Fig. 2b), leaving the SCNP agglomeration largely unaffected. Above the $T_{cp}$ of SCNP$_2$, both components agglomerate together into larger particles with diameters of 400–800 nm, as shown in Fig. 2b.

The absorption behavior of the here presented SCNPs is mainly influenced by the single-chain collapse procedure. The addition of free pyrrole during the single-chain collapse reaction had a major influence on the photophysical properties of the resulting PPy inside the SCNPs. While

all reactions led to the formation of pyrrole-pyrrole bonds as proven by the successful synthesis of SCNPs, only reactions with additionally added pyrrole were able to form PPy chains of sufficient chain length to form useful chromophores for the subsequent photothermal excitation. Absorption spectra of the SCNPs displaying different contents of embedded PPy are shown in Fig. 3a. It is clearly visible that SCNP$_{10}$ is showing a strong absorption over the complete VIS/NIR-range, while the absorptions of especially SCNP$_0$ and SCNP$_2$ are low or negligible.

We assume that, due to the overall low absolute total quantity of pyrrole units during the oxidative polymerization, mostly oligo pyrroles instead of PPy are generated, which display a poor optical density, as indicated by the low mass extinction coefficient $\varepsilon$ calculated from the Lambert-Beer plots of the SCNPs as shown in Fig. 3b, Figure S10, and Table 1. A higher extinction coefficient $\varepsilon$ was observed when more pyrrole was added during the oxidative polymerization, with SCNP$_{10}$ exhibiting the highest value with 14.25 mL mg$^{-1}$ cm$^{-1}$ (at 815 nm) and 15.31 mL mg$^{-1}$ cm$^{-1}$ (at 722 nm), comparable in magnitude to other reported, soluble PPy-nanosystems[16,17]. Measurements of the absorption behavior in organic solvents (methanol, THF, chloroform) revealed more pronounced absorption maxima at 653 nm and 845 nm in less polar solvents, especially chloroform, as depicted in Figure S11. In none of the selected solvents fluorescence could be detected, which proves that the PPy based systems are well suitable for photothermal applications as the excitation energy is to a large extent transformed into heat.

The strong absorption of SCNP$_{10}$ in water (Fig. 3c) allowed its application in biomedical methods that involve photothermal energy conversion. Irradiation of solutions at various concentrations with 5 ns pulses of an 815 nm laser (5.5 W/cm²) resulted in efficient conversion of the irradiation energy into heat *via* the photothermal effect, as depicted in Fig. 4a–e. Solutions were irradiated for 10 min, the temperature was measured, and then allowed to cool to room temperature for 15 min before the next irradiation cycle was started. At the end of the first irradiation cycle, temperature variations of 32.9 K at 0.1 mg/mL to 56.8 K at 2 mg/mL were

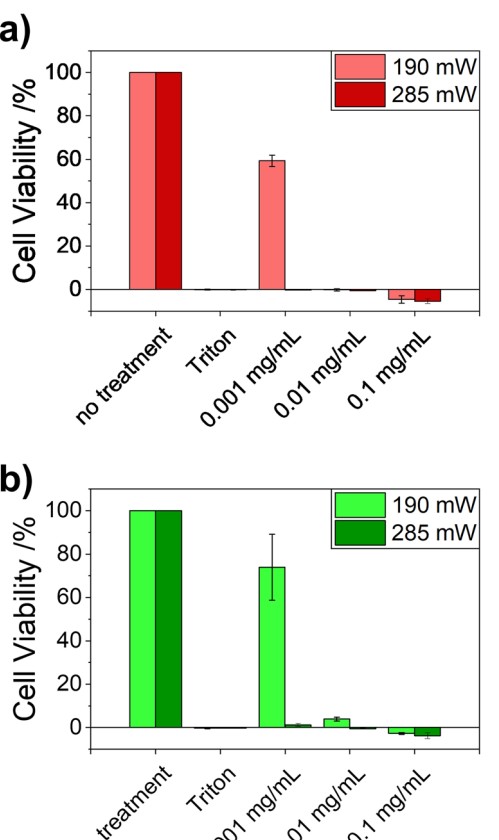

**Fig. 6 | Photothermal cell viability assays of SCNP$_{10}$. a** 3T3 and (**b**) DLD-1 cell lines after 15 min irradiation using the well plate NIR-LED illumination mat (722 nm) with radiant fluxes of 190 mW (0.56 W/cm) and 285 mW (0.84 W/cm²) and different concentrations of SCNP$_{10}$. Viabilities were measured after 96 h of incubation after the irradiation. Measurements without SCNP$_{10}$ and with only Triton were used as 100% and 0% references, respectively.

observed (see Fig. 4f and Table 2). Even though the SCNP structures reported here are significantly smaller than previously reported PPy-based materials, the temperature effects are comparable to other structures, with the fully soluble nature of the chromophore as a major advantage[16,17,19].

We further observed a direct transfer of the generated heat upon photoexcitation to the outer shell of the SCNPs by triggering its LCST behavior: photoexcitation and heat generation increased the local temperatures above the SCNP's cloud point temperature (see Fig. 2) causing the SCNPs to aggregate, as depicted in Fig. 4g–i. This process was largely reversible upon cooling to room temperature. Because the lag-times (15 min) in between the irradiation cycles were not sufficiently long for a quantitative redissolving of all SCNPs (see Fig. 4i), slight reductions of the absorption of the solutions and, hence, their photothermal activity were observed. Nevertheless, this photo-thermoresponsive effect is attractive, as it displays a unique behavior for the desired application in photothermal therapy. It is envisaged for a self-amplifying photothermal therapy approach, where at the beginning of a treatment with photothermal therapy, small concentrations of the SCNPs in the targeted tissue heat their surrounding above $T_{cp}$, in turn leading to agglomeration and precipitation of more SCNPs in the irradiated region. This then can increase the total concentration of PPy in the targeted tissue, increasing the photothermal activity for heat-induced cell deaths. Since after stopping the irradiation the SCNPs redissolve upon cooling, they can be easily removed from the body through renal clearance, because of their small size below 10 nm.

To test if similar photothermal energy conversions will be achieved under conditions relevant for living systems, irradiation experiments were conducted using lower light intensities with an NIR-LED array (722 nm) for

96 well plates. Each well in the 96-well plate was irradiated by a single LED with a radiant flux of 285 mW per well resulting in an irradiance of 0.84 W/cm². The temperature changes were followed using a thermal camera, as depicted in Fig. 5.

Even at such low light intensities a temperature increase from 15.9 °C to 63.0 °C ($T_{var}$ = 47.1 °C) was observed after 15 min of irradiation, proving the strong efficiency of the here reported SCNPs with light into heat efficiencies of up to ~75% (see Table S1). As seen from the temperature increases in Fig. 5e, an increase of concentration does not necessarily lead to a higher energy conversion, as already lower concentrations lead to an almost complete absorption at the bottom of the well and hence the upper regions of the well cannot contribute to the photothermal conversion.

The suitability of the here presented SCNPs for medical applications in PTT and PAI is further underscored by their low toxicity, which was measured by cell viability tests on 3T3 and DLD-1 cell lines after 24 h and 96 h, respectively (see Figure S12) with a focus on SCNP$_0$ and SCNP$_2$ to avoid interference with the fluorescent dye used to determine cell viabilities (see method section in the supplementary information)[37].

The main medical application of photothermal heat generation is in photothermal therapy (PTT), in which the locally generated heat is used to kill tumor cells. SCNP$_{10}$ shows a high PTT effectivity, as shown in Fig. 6. Two cell lines, 3T3 as healthy fibroblast cell line and DLD-1 as exemplary cancer cell line, were treated with different concentrations of SCNP$_{10}$ and then illuminated with the well plate NIR-LED illumination mat (722 nm) with radiant fluxes of 190 mW (0.56 W/cm²) and 285 mW (0.84 W/cm²) for 15 min. The SCNP concentrations and radiant fluxes were chosen to show negligible cytotoxic effects by the shear presence of light or the SCNPs (see Figure S12 and S13). After replacement of the SCNP-containing medium and additional 96 h of incubation with fresh medium, cell viabilities were measured. At already low SCNP concentrations of 0.001 mg/mL, representing PPy concentrations of 0.2 μg/mL, complete cell deaths were observed in both cell lines, when the higher radiant flux was used. Using the lower radiant flux resulted in complete cell deaths when the SCNP concentrations were at 0.01 mg/mL (2.0 μg/mL PPy). In general it can be said, that all concentration and radiant flux combinations showed significant cytotoxic effects, which became stronger at increased concentrations and higher fluxes. While this is the aimed effect for the cancer cell line, it is disadvantageous for the healthy lines. Here, a tissue specific targeting is necessary, e.g., by indirectly targeting tumor tissue *via* the above mentioned EPR effect, or other methodologies, which will be addressed in future studies.

Next to the light-induced heat generation, the here presented SCNPs could also display an additional cytotoxic effect caused by their LCST-type thermoresponsivity, which can also be triggered by light irradiation as shown above. Heating of the SCNPs above their $T_{cp}$ in a complex mixture with the model protein BSA in phosphate buffered saline causes their agglomeration into larger particles including also the model protein BSA, as described above. Thus upon irradiation with NIR light, the SCNPs first generate heat, leading to heat induced cell damages, then agglomerate, and thereby additionally harm the cell by interacting with the cell-internal components, like proteins.

We further subjected the PPy-SCNPs to photoacoustic imaging (PAI), a novel imaging method, able to achieve high resolutions, which is based on photothermal excitation of chromophores[54]. The here presented SCNPs were also suitable as a contrast agent for this imaging technique. As depicted in Fig. 7, SCNP$_{10}$ was used to generate PA signals in a PA spectrometer upon irradiation with a pulsed 815 nm laser. The results in Fig. 7a show a linear dependency of PA amplitude with concentration, as expected for a pure absorber which is desirable for molecular PA imaging.

Next to the general suitability for the creation of PA signals, SCNP$_{10}$ displays a novel contrasting effect through its photo-thermoresponsivity as depicted in Fig. 7b–d. Two tubes in a tissue phantom were imaged with a SCNP$_{10}$ solution in one tube and a CuSO$_4$ solution in the other tube as reference (see Fig. 7 and S4). Two images were captured with a temporal distance of 20 min. The first image was taken using a heat-induced-agglomerated solution of SCNP$_{10}$. To ensure the sample reached the

**Fig. 7 | PPy-based SCNPs as photoacoustic contrast agent. a** PA amplitude measured in PA spectrometer as a function of $SCNP_{10}$ concentration suggests a linear relation. **b** Schematic illustration of the tissue phantom used for the PA imaging which contains two tubes filled with $SCNP_{10}$ as a contrast agent (top left) and $CuSO_4$ as a negative sample (bottom right). **c** The difference PA image between two successive acquisitions in homogenous $SCNP_{10}$ solutions demonstrating that both tubes are eliminated down to the noise level. **d** In the difference PA image between homogenized and agglomerated $SCNP_{10}$ still the contribution of $CuSO_4$ is removed. This image clearly shows the changes in $SCNP_{10}$ tube and provides a contrast mechanism that is specific to these nanoparticles.

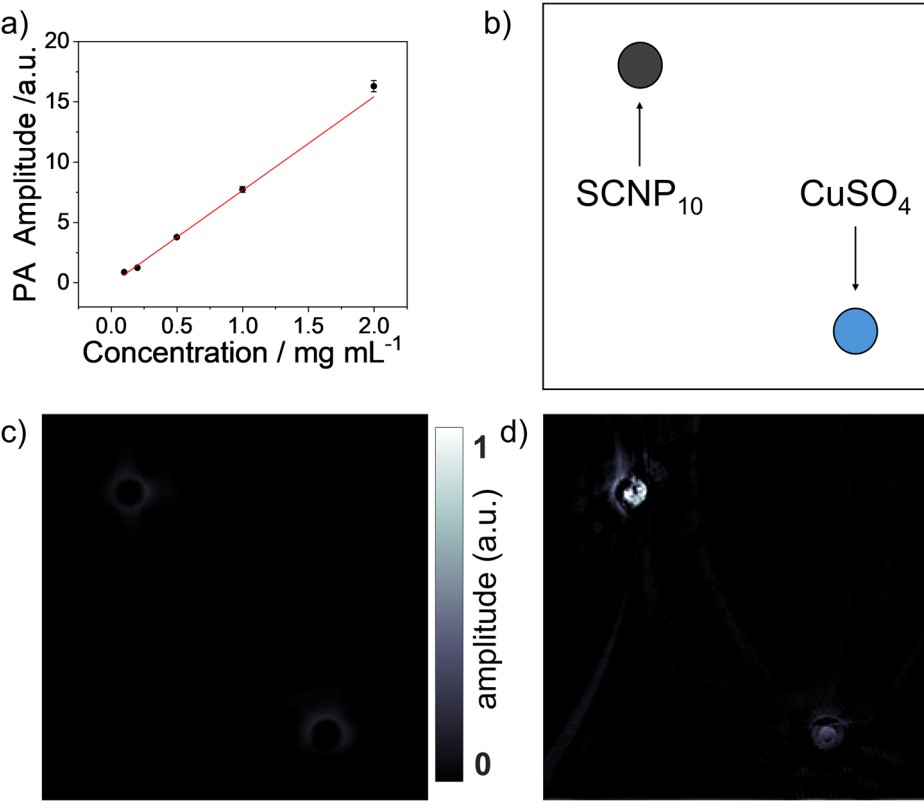

homogenized condition, the solution was kept inside the tomograph for 20 min and then the second PA image was captured. Since the PA efficiency only changed in $SCNP_{10}$ tube between these two acquisitions, the difference image (Fig. 7d) produced a clear contrast for the SCNP-filled tube, standing out from the tissue-background and the reference. To confirm that the contrast solely reflects variations in the homogeneity of the sample, a difference image was generated to compare two acquisitions with the tubes filled with homogenized SCNPs, as shown in Fig. 7c. The results indicate that any observed changes in both tubes are at the noise level.

## Conclusion

We successfully embedded polypyrrole (PPy) chromophores by in situ-polymerization into single-chain nanoparticles (SCNPs), which can be used as chromophores for photothermal effects in a dual approach for photothermal therapy and photoacoustic imaging. A set of biocompatible SCNPs was synthesized from an OEGMA-based precursor polymer with pyrrole-containing polymers, crosslinked directly via oxidative polymerization reaction inside the SCNPs, yielding SCNPs with sizes from 6.8 to 8.9 nm. PPy polymers can so be generated inside the SCNPs, with strong chromophores formed, surrounded by a hydrophilic and thermoresponsive SCNP-shell, rendering the normally insoluble PPy well water-soluble, still maintaining the strong absorption behavior of PPy-bulk material. Irradiation of aqueous solutions with a 815 nm laser or 722 nm LEDs induced temperature jumps $T_{var}$ of up to 56 °C, an effect which causes nearly complete cell deaths when cells were irradiated even at a low SCNP concentration (0.001 mg/mL) and low irradiance (0.84 W/cm²). This underlines the suitability of the here presented SCNPs for photothermal therapy and photoacoustic imaging, demonstrating that such small-sized nanoparticles (6.8 nm–8.9 nm) are efficient in photothermal conversion. The specific molecular design of the here presented nanoparticles, starting from polymer synthesis, the combination of the photothermal behavior of the PPy inside the SCNP together with its thermoresponsivity, resulted in a self-amplifying photothermal effect, further triggering a reversible agglomeration of the OEGMA-based SCNPs, also including biological entities like BSA, redissolving after the irradiation was stopped. We propose that this

unique photo-thermoresponsive effect opens novel terrain for photothermal therapy and photoacoustic imaging, as the heat induced enrichment of the SCNPs can result in a self-amplifying photothermal and photoacoustic effect, gradually enhancing the efficiency in and around the irradiated tissue. Together with the knowingly deep-reaching effect of the NIR light several centimeters inside tissue, this can help to push forward the benefits of photothermal therapies.

## Methods

*Analytical and Synthetic Methods* are described in the Supplementary Information.

### Reporting summary

Further information on research design is available in the Nature Portfolio Reporting Summary linked to this article.

## Data availability

The evaluated data of this study are available within this article and its Supplementary Information. The raw data that support the findings of this study are provided to the publisher and are additionally stored electronically (eLabFTW) according to the requirements of the DFG.

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

## Acknowledgements

We thank the DFG project BI1337/17-1, the GRK-2670, BEAM initiative (DFG, TP B01/Binder) and the "PoliFaces" initiative for financial support.

## Author contributions

Conceptualization, planning, and writing of this study were done by J.F.T. and W.H.B. J.F.T. was further responsible for the syntheses, chemical and structural analyses, photophysical characterizations, and LED illumination experiments. D.S., F.G.G., and J.L. were responsible for photothermal and photoacoustic measurements/imaging and their evaluation. H.L., J.K., and K.M. performed and evaluated the cell viability tests. All authors were involved in the reviewing process of this publication.

## Funding

## Competing interests

The authors declare no competing interests.
