## [Peer Review file · Communications Chemistry]

Photo-Thermoresponsive Polypyrrole-Crosslinked Single-Chain Nanoparticles for Photothermal Therapy

Corresponding Author: Professor Wolfgang Binder

Version 0:

Reviewer comments:

Reviewer #1

(Remarks to the Author)

This paper by Binder and coworkers provides a new way to produce functional single-chain nanoparticles useful for photothermal therapy. The synthetic approach to these versatile nanoparticles is totally new and involves the use of appropriate copolymer precursors combined with the external addition of pyrrole monomer to tune the photothermal properties. The main conclusions are supported by the experimental results, but the work requires some minor improvement before publication:

- 1) Abstract is totally absent in the present version of the manuscript.
- 2) Results provided in Figure 1c are not consistent with $T_{c,p}$ values reported in Table 1, at least for sample SCNP10.
- 3) A typo in line 85 "...SCNP vy additionally..." must be replaced by "...SCNP by additionally..."

Reviewer #2

(Remarks to the Author)

Authors demonstrated polymer nanoparticles based on polypyrrole possessing photothermal abilities. The manuscript is not well-designed and the conclusions are lack of data support especially the part of PTT. I don't think this work could be published at present stage. There are some comments below:

1. The Abstract is missing. Please add it
2. There are no data to support the potential of PTT in cells or animals. Authors should test the PTT abilities of nanoparticles in real cells and animals
3. The PA imaging data are rough. Please give some real figures and quantitative relation between PA signal and the concentration of PPs-SCNPs
4. The laser power is 5.5W/cm². It is super high. Without SCNPs, the temperature of water would also increase. The photothermal conversion efficiency of SCNPs should be calculated.
5. Pleas adjust the laser power so the light could be used for cell and animal experiments.
6. There are wrong data (sudden fall and rise) in figure 3a. please check.

Reviewer #3

(Remarks to the Author)

PTT is a promising method for cancer treatment, and photothermal systems are a hot topic in medicinal chemistry. The topic of the manuscript is interesting and suitable for publication in a communication chemistry journal. However, several points could be addressed to improve the quality of the manuscript.

The spectral properties of SCNPs require a more comprehensive description. For instance, it is essential to include their maximum absorbance values. Additionally, the absorption spectra of SCNP10 could suggest potential aggregation behaviour in aqueous systems. It would also be beneficial to present its spectra in both polar and nonpolar organic solvents.

Furthermore, Table 2 should include values for the other SCNPs, at a minimum in SI.

In addition to cytotoxicity, the phototoxicity of SCNPs should also be reported to provide a more complete understanding of their biological effects.

The laser-induced aggregation can be significantly influenced by the presence of biopolymers, such as proteins. Therefore, a model study involving HSA (most abundant human serum proteins) should be conducted to assess this interaction.

It is also pertinent to inquire whether SCNPs exhibit fluorescence properties, as this could have implications for their application in imaging or therapeutic contexts.

Table 1 presented molecular weight of SCNPs is significantly below 20 kDa. In this case of polymer systems, such as proteins higher accumulation in tumour tissue is typically associated with molecular weights of 20 kDa and above. Thus, a more thorough discussion regarding the possible accumulation of SCNPs in target tissues is warranted.

How are mass/volume PTT aggregates? Could SCNPs aggregation itself display toxic effects against cells?

Version 1:

Reviewer comments:

Reviewer #1

(Remarks to the Author)

In its present, revised and extended version the manuscript is ready for publication in Communications chemistry.

Reviewer #2

(Remarks to the Author)

This work can be published without any revision.

Reviewer #3

(Remarks to the Author)

The quality of the manuscript has been significantly improved. I have no serious objections.

Reviewer #1 (Remarks to the Author):

This paper by Binder and coworkers provides a new way to produce functional single-chain nanoparticles useful for photothermal therapy. The synthetic approach to these versatile nanoparticles is totally new and involves the use of appropriate copolymer precursors combined with the external addition of pyrrole monomer to tune the photothermal properties. The main conclusions are supported by the experimental results, but the work requires some minor improvement before publication:

1) *Abstract is totally absent in the present version of the manuscript.*

Answer: We added an abstract to the manuscript – sorry for the inconvenience.

2) *Results provided in Figure 1c are not consistent with T_{cp} values reported in Table 1, at least for sample SCNP10.*

Answer: We reevaluated the turbidimetry data and noticed that the T_{cp} values presented in Table 1 are in fact matching, contrary to the reviewers comment. The graphical representation in Figure 1c might have caused some confusion, and we adapted the figure for a better data representation.

3) *A typo in line 85 "...SCNP vy additionally..." must be replaced by "...SCNP by additionally..."*

Answer: We checked the complete manuscript again for any additional spelling mistakes.

Reviewer #2 (Remarks to the Author):

Authors demonstrated polymer nanoparticles based on polypyrrole possessing photothermal abilities. The manuscript is not well-designed and the conclusions are lack of data support especially the part of PTT. I don't think this work could be published at present stage. There are some comments below:

1. *The Abstract is missing. Please add it*

Answer: We added an abstract to the manuscript – sorry for the inconvenience.

2. *There are no data to support the potential of PTT in cells or animals. Authors should test the PTT abilities of nanoparticles in real cells and animals*

Answer: As a follow up of this comment, we measured the suitability of our SCNPs for PTT by irradiating two cell cultures (3T3 as healthy and DLD-1 as cancer cell cultures) using an LED array for the specific radiation of 96-well plates. We surprisingly found high effectivities in the light induced cell deaths – data are now included. For ethical reasons, we cannot conduct animal testing in the stage of our research, but would like to underscore that the now established 96-well format allows an easy, and also statistically significant parallelized testing under conditions closely meeting those of an *in-vivo* approach. We so believe that this model allows to prove the efficiency of our PPY-SCNP system.

3. The PA imaging data are rough. Please give some real figures and quantitative relation between PA signal and the concentration of PPs-SCNPs

Answer: Thanks to the reviewers comment we intensified our PA analyses. Next to the requested linear dependence of the PA signal and the concentration, we found a novel contrasting mechanism, based on the proposed photo-thermoresponsivity of the PPy-based SCNPs. We added the new data into the manuscript.

4. The laser power is 5.5W/cm². It is super high. Without SCNPs, the temperature of water would also increase. The photothermal conversion efficiency of SCNPs should be calculated.

5. Pleas adjust the laser power so the light could be used for cell and animal experiments.

Answer to 4 and 5: This laser power is indeed far above the clinically allowed power. We anyway decided to keep the results in the main text, since they show the interesting photo-thermoresponsive effect. Instead of new measurements using a weaker laser, we switched to an LED lamp with much lower irradiances (< 1 W/cm²) in the range of clinically allowed powers. There we also found high conversion efficiencies of up to 76%, again leading to high temperature increases. We additionally tested if this new setup would heat up water and indeed found a certain increase in temperature without the SCNPs inside. This slight temperature increase can be attributed to the warming of the LED lens matt during the irradiation and the thermal radiation which was coming from the other samples (see Figure 5 in the main text).

6. There are wrong data (sudden fall and rise) in figure 3a. please check.

Answer: The sudden fall and rise in temperature was probably caused by gas bubbles or precipitations of SCNPs on the thermal probe. We excluded the questionable data points from the graph.

Reviewer #3 (Remarks to the Author):

PTT is a promising method for cancer treatment, and photothermal systems are a hot topic in medicinal chemistry. The topic of the manuscript is interesting and suitable for publication in a communication chemistry journal. However, several points could be addressed to improve the quality of the manuscript.

The spectral properties of SCNPs require a more comprehensive description. For instance, it is essential to include their maximum absorbance values. Additionally, the absorption spectra of SCNP10 could suggest potential aggregation behavior in aqueous systems. It would also be beneficial to present its spectra in both polar and nonpolar organic solvents.

Answer: So far, in aqueous solutions, there was no actual absorption maximum observable, which is why we report the extinction coefficient at the laser excitation wavelength. We thank the suggestion of the reviewer, the absorption behavior in organic solvents (methanol, THF, chloroform) resulted in more pronounced absorption bands, enabling the detection of two major maxima. We added this information to the main text and added the spectra in the SI (Figure S11)

Furthermore, Table 2 should include values for the other SCNPs, at a minimum in SI.

Answer: Because of the negligible absorption of the other SCNPs, there is no detectable photothermal effect.

In addition to cytotoxicity, the phototoxicity of SCNPs should also be reported to provide a more complete understanding of their biological effects.

Answer: Using our LED array setup for the irradiation of cell cultures we measured the phototoxicity of the here used NIR light and found only minor effects on the cell viabilities. We added this information to the main text and supporting information.

The laser-induced aggregation can be significantly influenced by the presence of biopolymers, such as proteins. Therefore, a model study involving HSA (most abundant human serum proteins) should be conducted to assess this interaction.

Answer: The presence of biopolymers indeed has an influence on the aggregation behavior. We performed temperature dependent DLS measurements of SCNP₂ in presence of BSA as widely used model protein. We found two results: (i) BSA agglomerates together with the SCNPs above 50°C while showing only minor changes in D_h without the SCNPs; (ii) the aggregates that the SCNPs form with BSA above 50°C are significantly smaller than the agglomerates of only SCNP₂.

It is also pertinent to inquire whether SCNPs exhibit fluorescence properties, as this could have implications for their application in imaging or therapeutic contexts.

Answer: We tested the fluorescence behavior of the SCNPs in water and the previously mentioned organic solvents with excitations at 500 nm, 600 nm, 700 nm, and 800 nm. The fluorescence measurements resulted in no measureable emission in the range up to 1000 nm.

Table 1 presented molecular weight of SCNPs is significantly below 20 kDa. In this case of polymer systems, such as proteins higher accumulation in tumour tissue is typically associated with molecular weights of 20 kDa and above. Thus, a more thorough discussion regarding the possible accumulation of SCNPs in target tissues is warranted.

Answer: In literature (e.g. Langer et al. *Nature Nanotechnology* **2007**, Mitragotri et al. *Journal of Controlled Release* **2024**, citations now added to the manuscript) different nanoparticles in combinations with different cancer types have been described, with nanoparticle sizes ranging from two to several hundred nanometers. There is not a clear indication which size of the nanoparticles are advantageous, but it is rather case dependent. Exemplary, Liang et al. (DOI: 10.1021/nn301282m) found a better accumulation of 2 nm sized nanoparticles in MCF-7 cells, compared to the 15 nm sized counterparts, so that it is unclear which size would be optimal, also considering that the surface of such NPs might be even more important than the size-selective uptake. Since the sizes of nanoparticles for triggering the EPR effect are of great relevance, as the reviewer correctly implied, we added a more detailed description of the EPR effect of nanoparticles in the manuscript.

How are mass/volume PTT aggregates? Could SCNP aggregation itself display toxic effects against cells?

Answer: To address this question we performed temperature dependent DLS measurements of SCNP₂ together with BSA as widely used model protein. We found that BSA agglomerates together with the SCNPs above 50°C, resulting in agglomerates of ~400-600 nm in size overall. Thus the agglomeration of such SCNPs (under more realistic conditions also including albumin-proteins) could create an additional toxic effect in the NIR-irradiated tissue, which goes beyond the originally intended photothermal cell death.